# Enhanced Photocatalytic Activity of Nonuniformly Nitrogen-Doped Nb_2_O_5_ by Prolonging the Lifetime of Photogenerated Holes

**DOI:** 10.3390/nano12101690

**Published:** 2022-05-16

**Authors:** Wei Guo, Chang Bo, Wenjing Li, Zhiying Feng, Erli Cong, Lijuan Yang, Libin Yang

**Affiliations:** Tianjin Key Laboratory of Brine Chemical Engineering and Resource Eco-Utilization, College of Chemical Engineering and Materials Science, Tianjin University of Science and Technology, Tianjin 300457, China; weiguo@mail.tust.edu.cn (W.G.); bochang@mail.tust.edu.cn (C.B.); lwj0606s@mail.tust.edu.cn (W.L.); zhing@mail.tust.edu.cn (Z.F.); cel@mail.tust.edu.cn (E.C.); yanglijuan@mail.tust.edu.cn (L.Y.)

**Keywords:** photocatalysis, niobium pentoxide, photogenerated holes, degradation, 2,4-dichlorophenol

## Abstract

The narrow band gap and significant separation of photogenerated carriers are essential aspects in practical photocatalytic applications. Nitrogen doping usually narrows the band gap of semiconductor oxides, and it enhances photocatalytic activity. Nitrogen-doped Nb_2_O_5_ was prepared by a multiple hydrothermal method. The non-metal element N inside the nanostructure, working as the trapping sites for the holes, which were effectively incorporated into the crystal lattice of Nb_2_O_5_ semiconductor oxide, remarkably shorten the band gap (3.1 eV) to enhance the visible light response, effectively reducing the photoinduced electron–hole pair recombination and prolonging carrier lifetime. The multilayer coating structure with a gradient concentration distribution and the type of nitrogen doped is favorable for the migration of photoexcited carriers in the bulk of catalysts. The unique multi-layer coating with the micro-concentration gradient of doped nitrogen provides a fast separation channel and jump steps for the separation of electron–hole pairs.

## 1. Introduction

At present, the rapid consumption of fossil fuels and environmental pollution have been recognized by the whole world as two serious problems that need to be urgently solved. Researchers focused their efforts on cost-effective and sustainable semiconductor photocatalytic materials for the utilization of inexhaustible solar light. It is generally accepted that the photocatalytic mechanism involved in nanostructured semiconductor materials implies a complex sequence of photophysical and electrocatalytic processes: (i) the absorption of a photon with energy greater than the band gap causes the generation of electron (e^−^) and hole (h^+^) pairs in the bulk of photocatalysts, (ii) the separation and migration of photoexcited carriers or their recombination in the bulk or surface of catalysts, and (iii) the photo-redox reaction at the active site on the surface of catalysts [1,2]. The separation of photoinduced carriers depicts a crucial step to determine the efficiency of the solar energy conversion in photocatalytic reactions. To promote the separation of photoexcited carriers, a systematic heterogeneous catalyst consisting of multiple components, such as metal tips, was successfully obtained to achieve the physical separation of the redox active sites through various spatial arrangements underlying photophysical and photochemical phenomena. Various strategies were reported to gain long-lived charge-separating capabilities, such a given morphology and selected size [3], metal doping [4], and the introduction of quantum dots [5,6]. Noble metal nanoparticles, such as Au [4], Pt [7,8], or Pd [9], were dispersed on the surface of semiconductor materials as co-catalysts, and the migration of the photogenerated electron from the bulk phase to the reaction interface was particularly accelerated. These co-catalysts, working as electron traps, form the reduction reaction sites in the photocatalytic process.

In photocatalysis, the oxidation reaction is usually considered as a rate-determining step because of the slow rate of photogenerated holes migrated to surfaces and the high overpotential compared with the photoelectrons [10]. It is inevitable that photogenerated holes aggregate the trap states, to be recombined by the electrons or to continue hopping up to the surface of the catalysts to participate in the oxidation reaction in the migration. Among them, the recombination rate of photogenerated holes with electrons in the bulk of catalysts is in the range of a femtosecond to a picosecond of time, which greatly shortened the lifetime of carriers and limited utilization [11]. To restrict recombination, two approaches are usually recommended, one in which photoexcited holes are captured and consumed by a sacrificial agent [12,13]. Alternatively, the photoexcited holes are confined in a position of the catalyst as traps [14,15,16,17]. Meanwhile, the formation mechanism of hole traps in nonuniform lattices of catalysts has not been clearly discussed.

It is an expensive technique to adopt noble-metal loading on the surface of a semiconductor oxide, and it has an apparent disadvantage that clusters of metal as reduction sites can easily form states of recombination [18,19]. A great deal of effort has been focused on the adoption of a cheaper strategy of the application of nonmetallic elements modified with transition metal oxides, which would not only be conducive to shortening the band gap to expand the photo response range from the ultraviolet to visible light area, but would also inject impurity levels to boost the separation of photogenerated carriers [20,21,22]. The substitution of a N atom was preferentially considered as a result of its similar atomic radius to an oxygen atom and its p orbits contribute to the band-gap narrowing by mixing with O 2p orbits [23,24,25,26]. Due to the high stability of niobium pentoxide (Nb_2_O_5_) and its narrow band gap (about 3.4 eV), similar to TiO_2_, efforts are particularly important to explore photocatalysts with narrow band gap to promote the utilization of sunlight. Kulkarni et al. [27] exhibited Nb_2_O_5−x_N_x_ in photocatalytic activity towards hydrogen evolution for water splitting and for H_2_S splitting under visible light. Hu et al. [28] reported N-Nb_2_O_5_ nanobelt quasi-arrays for dye degradation. In addition, Lian et al. explored orthorhombic Nb_2_O_5_ nanoparticles on/in N-doped carbon hollow tubes for Li-ion hybrid supercapacitors [29]. However, the effects of the substitution or doping of N atoms into the lattices of Nb_2_O_5_ on photocatalytic activity, especially the migration of the photocarriers, remain debatable.

In this work, nonuniform N-doped Nb_2_O_5_ microspheres were prepared by the hydrothermal method. N atoms, replacing some lattice oxygen atoms in Nb_2_O_5_, enhanced the local delocalization of unpaired electrons and formed trapping sites for the photogenerated carriers (especially holes) to inhibit the recombination of the photoexcited carriers.

## 2. Materials and Methods

### 2.1. Materials

All the reagents were of analytical grade and they were used as received, without further purification. Niobium oxalate hydrate (C_10_H_5_NbO_20_·xH_2_O, 99.99%) was purchased from Heowns Biochemical Technology Co., Ltd., Tianjin, China, and it was as niobium precursor. Urea (H_2_NCONH_2_, ≥99.0%) was supplied by Vector Chemical Products Trading Co., Ltd., Tianjin, China, and it served as nitrogen source. In addition, cetyltrimethylammonium bromide (CTAB, C_19_H_42_BrN, ≥99.0%) and polyvinylpyrrolidone K30 (PVP, (C_6_H_9_NO)n, 44,000–54,000) were used as soft template and dispersant, respectively, and they were supplied by Sinopharm Chemical Reagent Co., Ltd., Tianjin, China. Sodium sulfate (Na_2_SO_4_, A.R.) and ethanol (C_2_H_6_O, A.R.) were purchased from Aladdin, Shanghai, China.

### 2.2. Synthesis of N-Nb_2_O_5_ Microspheres

The preparation of N-Nb_2_O_5_ microspheres is based on modified hydrothermal method [30]. As described in Figure 1, in a typical synthesis, to the solution of 0.2 g of CTAB, 1 g C_10_H_5_NbO_20_.xH_2_O, and 0.558 g urea (n_Nb_:n_N_ = 1:10) in ethanol and deionized water (V_ethanol_:V_water_ = 1:1, 60 mL) were slowly added with constant stirring at room temperature until a homogenous solution was achieved. Subsequently, the transparent mixture was transferred to a 200 mL Teflon-lined stainless-steel autoclave, heated at 180 °C for 24 h and then cooled at room temperature naturally. Afterwards, the precipitates were centrifugated, washed with deionized water and ethanol, and then dried overnight at 110 °C. The obtained powder, heated at 600 °C for 3 h in air with a ramp of 5 °C min^−1^ in air atmosphere in a muffle furnace, is denoted as 10NNbO. In contrast, 2 NNbO (n_Nb_:n_N_ = 1:2) was synthesized by reduction of N source in the hydrothermal process. In addition, the pure Nb_2_O_5_ microsphere, as control sample, were fabricated in the absence of N source. 

The heterogeneous nitrogen-doped Nb_2_O_5_ microspheres were synthesized by a cyclic and multiple hydrothermal method (Figure 1). To control the distribution of N-doped concentration, the ratio of N source to niobium source and the concentration of the solution were varied in the hydrothermal synthesis. The prepared N-doped Nb_2_O_5_ microspheres as seeds were evenly dispersed in the solution with added PVP, while the other steps were repeated. The detailed procedure was as follows: 0.2 g of Nb_2_O_5_ or 2NNbO or 10NNbO seeds and 0.1 g of PVP were sequentially added into a mixture of 30 mL deionized water and 30 mL of ethanol in a cylindrical beaker. In the subsequent hydrothermal process, C_10_H_5_NbO_20_·xH_2_O and urea were added to the suspension according to the molar ratio of 1:5, 1:8 and 1:10, respectively, for sample 10@8@5@2NNbO. For sample 2@5@8@10NNbO, niobium source and nitrogen source were added to the suspension according to the molar ratio of 8:1, 5:1 and 2:1, respectively. For the samples Nb_2_O_5_@10NNbO, 10NNbO@ Nb_2_O_5_, the molar ratios of niobium and nitrogen sources were 1:10 and 10:1, respectively. Subsequently, the as-prepared solution was transferred to a 100 mL Teflon-lined autoclave, subjected to a hydrothermal process at 180 °C for 24 h and cooled naturally to room temperature. The resulting precipitate was centrifugated, washed and dried at 110 °C. The final sample was obtained via calcination at 600 °C for 3 h with a 5 °C/min heating rate. 

### 2.3. Photocatalytic Reaction

The photocatalytic activity was evaluated by the degradation of 2,4-dichlorophenol (2,4-DCP) (100 mg/L) under ultraviolet irradiation. In addition, an irradiation by a Xe lamp (300 W) with a UV cut-off filter (>365 nm) was adopted as an ultraviolet light source in a 300 mL quartz glass reaction vessel, with a flow of cool water to maintain the temperature at 25 °C during the reaction. A total of 50 mg of as-prepared sample was added into 50 mL of 2,4-DCP solution in a quartz vessel and treated by ultrasonic waves for 1 min. The suspension was stirred with magnetic stirrer in the dark for 1 h to reach the adsorption equilibrium states on the surface of the catalyst. Subsequently, the solution was irradiated by ultraviolet light for 180 min. A small amount of liquid (about 1 mL) was taken through 0.45 μm millipore membrane filter at regular intervals (0, 20, 40 90, 150 and 180 min). The external standard curve of low concentration of 2,4-DCP solution in water was determined with the use of a Shimadzu UV–vis spectrophotometer (UV-2700 PC) in 10 mm quartz cuvettes. The calibration curve was plotted to permit the extrapolation of the 2,4-DCP. The degradation efficiency of 2,4-DCP was calculated using Equation (1), where DE is the degradation efficiency of the degradation materials 2,4-DCP (%), C_0_ (mg L^−1^) is the initial concentration of 2,4-DCP and C_t_ (mg L^−1^) is the concentration of 2,4-DCP at irradiation time t (min).
(1)$$\mathrm{DE}=\left(1-\frac{C_{t}}{C_{0}}\right)\times 100\%,$$

### 2.4. Characterization of the Catalysts

The morphology, composition and microstructure of the as-synthesized catalysts were examined by a field emission scanning electron microscope (FE-SEM, Apreo S LoVac, FEI, Hillsboro, OR, USA) and transmission electron microscopy (TEM, JEM F200, JEOL, Tokyo, Japan) equipped with an energy dispersive X-ray spectrometer (EDS) operated at 200 kV. The crystalline phases in the prepared samples were characterized by an X-ray diffractometer (XRD-6100, Shimadzu, Tokyo, Japan) at 36 kV, 30 mA using Cu Ka X-rays. X-ray photoelectron spectroscopy (XPS, K-Alpha+, Thermo Fisher, Waltham, MA, USA) measurements were performed with a monochromatic Al Ka X-ray source to obtain the elemental composition and valence states, and all the binding energies were determined in reference to the C 1 s peak of 284.8 eV as the internal standard. The room-temperature Raman analysis of samples was performed using an inVia Reflex Raman spectrometer (HR-800, HORIBA Jobin Yvon, Paris, France) with 532.5 nm laser excitation. The UV-Vis diffuse reflectance absorbance spectra (DRS) were obtained with a UV-Vis-NTR spectrophotometer (UV-2700, Shimadzu, Tokyo, Japan) equipped with an integrated sphere and scanning in the range of 200 and 800 nm, while BaSO_4_ was used as a reference. The corresponding energy gap can be obtained according to a plot of the transformed Kubelka–Munk function versus the energy of the exciting light, as (αhυ)^1/2^ = A (hυ − Eg), where A is a constant, Eg is the band gap energy, α is optical absorption coefficient, and hυ is photon energy. The surface chemistry of the photocatalysts was analyzed by Fourier-transform infrared spectroscopy (FTIR, Bruker TENSOR27, Saarbrucken, Germany) in the transmission mode from 400 to 4000 cm^−1^. Steady fluorescence emission spectra were recorded at room temperature with a fluorescence spectrophotometer (FLS1000, Edinburgh Instruments Ltd., Scotland, Britain). The photoelectrochemical measurements were carried out on a workstation (CHI760E, Shanghai, China) in a three-electrode system consisting of photocatalyst-coated in a 1.5 cm × 2.5 cm ITO conductive glass substrate, platinum plate (20 × 20 mm), and saturated Ag/AgCl electrode (ϕ6 × 120 mm) as the working counter, and reference electrodes, respectively. For fabricating the working electrode, the sample powder (10 mg) was ultrasonicated in 1 mL of ethanol and dispersed evenly to form a homogeneous ink. Then, the suspension ink was spread on the ITO glass and then dried at 80 °C for 12 h. The working electrode was irradiated with a 300 W xenon lamp during the measurement, which was performed in 0.5 M Na_2_SO_4_ at room temperature. The illumination was interrupted every 30 s at a bias potential of 0.5 V vs. Ag/AgCl. 

### 2.5. Computational Methods and Models

All the calculations of the molecular and electronic structures for the photocatalysts were based on density functional theory (DFT) with the exchange–correlation functional at the generalized gradient approximation (GGA) level parameterized by Perdew–Burke–Ernzerhof (PBE) [31], as implemented in the Cambridge Serial Total Energy Package (CASTEP) codes, combined with ultrasoft pseudopotentials (USPP) [32]. The calculations were converged at a cut-off energy of 320 eV for the plane wave basis set, while the Brillouin zone (BZ) integrals were approximated using the special k-point girds with a mesh of 2 × 2 × 1 for geometry optimization and calculations of electronic properties, and the minimization algorithm was the Broyden–Fletcher–Goldfarb–Shanno (BFGS) scheme [33]. The T−Nb_2_O_5_ (space group Pbam) has an orthorhombic crystal structure with partial occupancy. To calculate the partial occupancy in theoretical simulations, a minimum of (1 × 1 × 5)-sized super cell would be needed. It contains 294 atoms with high computational power to perform the electronic structure analysis. In the lattice of T−Nb_2_O_5_, the oxygen atoms are coordinated to Nb atoms through distorted octahedral/pentagonal bipyramidal structures. To avoid the high computational cost of the calculation for the electronic structure, an orthorhombic structure model with Nb atoms in octahedral coordination and pentagonal bipyramidal coordination, as in the case of *β*−Ta_2_O_5_, was chosen. The geometry of N-doped Nb_2_O_5_ was re-optimized using a conjugate gradient algorithm [34] until the forces acting on each atom were converged below 0.03 eV Å^−1^.

## 3. Results and Discussion

### 3.1. Doping and Distribution of N in Nb_2_O_5_

The 10@8@5@2NNbO exhibits a typical microsphere morphology, as shown in Figure 1a, and the diameter of the rough microspheres is in the range of 0.5–2 μm. As shown in Figure 1b, a multi-layer structure is formed from the layered growth of a gradient precursor in the cyclic hydrothermal process. The high-resolution transmission electron microscopy (HRTEM) image of the area near the inner surface of Nb_2_O_5_ microsphere s (shown in Figure 1c) reveals the lattice spacing of 0.384 nm and 0.322 nm that are matched well with the planes (001) and (180) of the orthorhombic Nb_2_O_5_, respectively. Figure 1e–g clearly demonstrates that the elemental compositions of each layer of the catalyst are mainly based on Nb, O and N signals, being similar to that in the Figure 1d. A STEM-energy dispersive X-ray spectroscopy (EDS) line scan was performed through the center of an individual 10@8@5@2NNbO particle (shown in Figure 1h), which further confirms that the concentration of N doped decreases gradually from the edge to the internal core. Similarly, the microstructure and element concentration of 2@5@8@10NNbO, Nb_2_O_5_@10NNbO and 10NNbO@Nb_2_O_5_ are demonstrated in Appendix A. The nitrogen content was 3.01%, 2.24%, 1.94%, 2.19% for the samples of 10@8@5@2NNbO, 2@5@8@10NNbO, 10NNbO@Nb_2_O_5_ and Nb_2_O_5_@10NNbO samples, respectively. In Appendix A, it can be distinctly observed that the concentration of N gradually increases from the edge to the core, exactly opposite to the trend of 10@8@5@2NNbO. It results from the opposite concentration gradient of the N precursor during the preparation. The XRD patterns (Figure 1i) match well with the orthorhombic Nb_2_O_5_ (JCPDS No. 30-0873) with the diffraction peaks at 22.1°, 28.0°, 36.1°, 46.3°, 50.4°, 54.9°, indexed as (0 0 1), (1 8 0), (1 8 1), (1 1 4 0), (3 2 1) and (3 7 1) reflexes, respectively [35]. The absence of obvious peaks of the nitrogen element indicates that the introduction of a small amount of nitrogen does not damage and change the lattice arrangement. Meanwhile, the introduction of N causes the transition of the dominant crystal facet.

In Figure 2a, the broader absorption peaks near 1624 cm^−1^ were thought to be surface hydroxyl groups [36]. In addition, the characteristic absorption peak of as-prepared catalyst around 592 cm^−1^ can be assigned to the Nb=O stretching and Nb–O–Nb angular vibrations, while the observed peak at 880 cm^−1^ occurred due to the asymmetric stretching of O–Nb–O bonds [35]. The peak value has a significant decrease at 592 due to the substitution of N for O in Nb-O-Nb. The peak strength decreases significantly at 592, which is an excellent proof that the bridge oxygen in Nb–O–Nb was replaced to form a new Nb–N–Nb bond due to N. In addition, the emergence of the peak at 1420 cm^−1^ indicates the formation of NO_X_ species in the N-doped Nb_2_O_5_ microspheres [37]. Similarly, as shown in Figure 2b, for Nb_2_O_5_ and N-doped Nb_2_O_5_ samples, Raman shifts at 235 and 320 cm^−1^ are attributed to angle deformation modes and a bridge of Nb–O–Nb bonds, and peaks around 696 cm^−1^ assigned to the stretching vibration of a Nb–O bond [38]. The blue shift of the vibration peaks originally contributed to Nb–O–Nb, indicating the partial symmetry breaking caused by the introduction of nitrogen into the lattice. 

As shown in Figure 3a, the two strong peaks at 207.1 and 209.8 eV belong to the Nb 3d_5/2_ and Nb 3d_3/2_ of Nb (V), respectively [39,40]. Notably, the shift in the binding energies (B.E.) with respect to pristine Nb_2_O_5_ (207.6 and 210.3 eV) is attributed to the change in electron density around the Nb atom as a result of the substitution of an O atom with the less-electronegative N element [27,35,41]. The O1s core level spectrum in Figure 3b could be fitted with two peaks which are characteristics of a Nb-O bond (529.9 eV) as oxygen in hydroxyl groups (531.3 eV) [38,42]. Figure 3c,d depicts the N 1s core level spectrum, deconvoluted into two peaks 395.3 and 399.9 eV, which are assigned to the substitutional β-N in the metal oxide lattice and the interstitial γ-N in the Nb–O–N bond, indicating the presence of oxygen and nitrogen in the same lattice units [43].

### 3.2. Photo Response and Photodegradation Performance

The PL emission spectra of the as-prepared catalysts were monitored at the excitation wavelength 320 nm as the band edge emission, and the curves are shown in Figure 4a. Two distinct main peaks at 400 nm and 410 nm are assigned to the band–band transition of N-Nb_2_O_5_ and the formation of an oxygen defect, respectively [35]. The intensity of the PL emission for N-doped Nb_2_O_5_ is much lower than that of pristine Nb_2_O_5_, and this indicates that the recombination of photogenerated carriers is greatly inhibited by the introduction of N element. The sample 10@8@5@2NNbO exhibited the lowest intensity of PL emissions in all the synthesized photocatalysts, due to the special gradient distribution of N element. The substitution of N not only reduces the formation of oxygen defects, but also restricts the photogenerated holes and provides enough sites to accelerate hole hopping. To further evaluate the electron transport behavior, photocurrent response tests were carried out to investigate the charge separation efficiency. As shown in Figure 4b, the maximum photocurrent density (1.20 μA cm^−2^) produced by the electrode coated with 10@8@5@2NNbO is about four times that of pure Nb_2_O_5_ (0.34 μA cm^−2^). Under the same test conditions, the photocurrent responses of 2@5@8@10NNbO, 10NNbO@ Nb_2_O_5_, Nb_2_O_5_@10NNbO were 1.87 μA cm^−2^, 1.02 μA cm^−2^ and 0.40 μA_·_cm^−2^, respectively. Interestingly, the extremely identical signal values of Nb_2_O_5_@10NNbO and Nb_2_O_5_ indicated that the migration of the photoelectron is also severely inhibited when the N-doped Nb_2_O_5_ as core was coated by an Nb_2_O_5_ crystalline outer layer. Such response amelioration caused by nitrogen doping could be attributed to the combined effect of morphology and an enhanced light absorption threshold to the lower energy end. In Figure 4c, pure Nb_2_O_5_ has edge absorption at about 380 nm, whereas other the N-doped materials have a significant red shift from 380 to 400 nm. The improved absorption for the N-doped samples is ascribed to the fact that the additional electronic states hybridized by the N_2p_ orbital are located above the valence band derived mainly from O_2p_ orbitals. The band gap energy (Eg) for the 10@8@5@2NNbO, 2@5@8@10NNbO, 10NNbO@ Nb_2_O_5_, Nb_2_O_5_@10NNbO and Nb_2_O_5_ is equal to 3.22, 3.25, 3.10, 3.18 and 3.35 eV, respectively. The introduction of N as the recombination center inhibiting the separation of electron and hole pair, thus corresponding to the decrease in photocurrent density [44], and band gap width, does not change significantly. The widening trend of band gap width is consistent with the concentration of the doped N.

As an organic pesticide, 2,4-dichlorophenol(2,4-DCP) can effectively improve the yield of crops, but its excessive use causes accumulation in water and soil, and ecological pollution [45]. Usually, these contaminants can be degraded in under light radiation. The degradation process of 2,4-DCP is proposed by Equations (2)–(4) in UV-irradiated suspension. The degradation efficiency of 2,4-DCP in as-prepared catalysts, with adsorption equilibrium in darkness for 1h before ultraviolet radiation, is shown in Figure 4d and Appendix A to evaluate photocatalytic activity. The degradation efficiency of 2,4-DCP in 10@8@5@2NNbO, 2@5@8@10NNbO, 10NNbO@ Nb_2_O_5_, Nb_2_O_5_@10NNbO and Nb_2_O_5_ under UV radiation for 180 min reached 57.39%, 40.09%, 34.26%, 19.73% and 26.36%, respectively. The multilayer coated N-doped Nb_2_O_5_ microsphere showed higher activity for the degradation of 2,4-DCP than the double-layer coated structure under the same experimental conditions and N-doping amounts. The results of the photodegradation experiments proved that a multilayer coating structure with the multi-stage concentration distribution of N was more favorable for the separation and migration of photoexcited carriers than the double-layer structure. A gradient N doping provides the constraint sites to accelerate hole hopping at the catalyst surface, but a large span of N concentration in the microspheres forms crystalline defects, and it increases the recombination of photogenerated electrons and holes. When N concentration gradually increases from core to outside, the rapid migration of photogenerated carriers to the surface greatly inhibits the recombination of electron–hole pairs in the bulk to participate redox reaction. On the contrary, the slow migration of photogenerated carriers can easily be recombined with the defects in the bulk, to decrease the photocatalytic performance. Combined with the above XPS analysis, it was found that sample 10NNbO@ Nb_2_O_5_, comprised of more β-N structures, was more conducive to the separation of the bound holes from the local defect states in the bulk compared with the β-N structure in Nb_2_O_5_@10NNbO. The excellent catalytic performance of 10@8@5@2NNbO and 2@5@8@10NNbO may also be attributed to the interaction effects of both type N. Excessive γ-N causes the strong binding of holes and inhibits its further migration. It is particularly important that crystalline defects, as the active sites in the bulk of catalysts, play a crucial role in photocatalytic oxidation reactions.
N-Nb_2_O_5_ + hv → e^−^_CB_ + h^+^_VB_(2)
h^+^_VB_ + H_2_O → OH + H^+^(3)
OH + 2,4-DCP → 3,5-dichlorocatechol + 2,4-dichlorophen-1,5-diol(4)

### 3.3. Enhancement Mechanism of Photocatalytic Activity

To reveal the effect of nitrogen introduction on the migration of the photoexcited carriers, the electron density of the 10@8@5@2NNbO-modeled structure was calculated based on DFT theory. It is known that the introduction of N creates new impurity levels by mixing with O 2*p* states, which contributes to the elevation in the valence band top [23]. It is especially necessary to pay attention to the fact that interstitial N has a deep localized state, but, as a recombination center, greatly inhibits the separation of photogenerated electrons and holes [46]. Therefore, the formation of N directly substituted in the doping type is more conducive to the improvement of photocatalytic efficiency in the lattice. Figure 5a displays the dominant (001) plane model of N-Nb_2_O_5_, and Figure 5b is an electron concentration distribution diagram displaying that the N atoms replace the bridging oxygen and merge into the lattice to form a local electron-rich region. In the multilayer coating structure, N concentration is increased with a small gradient from the bulk to the subsurface, and it results in the local crystalline defects effectively weakening the potential barrier for the bound holes and accelerating the migration of the carriers from the bulk to the surface-active site. Figure 5c illustrates the accelerated migration process of photogenerated holes with the doped N concentration increases in the Nb_2_O_5_ crystalline. Due to the trapping effect [47], photogenerated holes in the bulk tend to escape the trap sites. When stimulated by the light irradiation, the driving force strengthened by the gradually deepening traps improves the migration process of holes from the bulk to the surface, and it is conducive to prolonging the lifetime of the holes and the separation from electrons. It can be concluded that the ordered crystalline lattice modified by heterogenous element constructed a directional channel for the migration of holes to prolong the lifetime and improve the utilization of light quantum in photocatalytic reactions [16,23,48].

## 4. Conclusions

Unique multilayer coated microspheres were synthesized by a multiple hydrothermal method by adjusting the concentration and ratio of niobium and N sources in the solution. An ordered gradient concentration of doped N locally substituted oxygen atoms was introduced into the orthorhombic niobium oxide lattice and characterized by XPS, TEM with EDS. The photocurrent measurements and steady-state fluorescence spectra demonstrated that the recombination of photogenerated electrons and holes greatly reduced in the Nb_2_O_5_ doped with an ordered gradient concentration of N. Compared with the double-layer coated microspheres with a larger concentration span, the multi-layer coated microspheres with a small gradient concentration were more conducive to the degradation of organic compounds by photogenerated holes in the oxidation reaction. As a result, with the gradual increase in N concentration from the core to the outside surface, the mechanism of the photocatalytic reaction in multilayer coated microspheres was tentatively proposed and was proven by the differential electron densities caused by gradient N doping. The formation of directional channels and ordered crystalline lattices modified by heterogenous element promoted the migration of the accumulated holes in the bulk to the surface to participate in photocatalytic reaction.

## Data Availability

Not applicable.

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
