# Peer review of "Enhanced Photocatalytic Activity of Nonuniformly Nitrogen-Doped Nb2O5 by Prolonging the Lifetime of Photogenerated Holes"

_nanomaterials, 2022, doi:10.3390/nano12101690_

Round 1

Reviewer 1 Report

Please check the attached PDF file for detail comments

Reviewer 2 Report

  1. In my opinion the most important results are not supported by the experimental data. The gradual distribution of nitrogen, concluded from EDS results, is disputable. As shown in Fig. 1H and Fig. S1-S3, signal typical of N is strongly correlated to the signal typical of Nb. The lower the signal typical of Nb, the lower the signal typical of N. The same in the opposite situation. The higher the signal typical of Nb, the higher the signal typical of N. It means that different distribution of nitrogen may result from different thickness of the analyzed sample/region. The thicker the sample (analyzed region), the more intense signal typical of both N and Nb.
  2. The authors claim that photo-generated holes play important role in the degradation process. However, they have not provided any evidence to support their hypothesis.
  3. Fig. 1i shows that samples treated hydrothermally several times were much more crystalline than the rest of materials. It is clear that higher crystallinity or larger crystallite size may have significant impact on photocatalytic activity of the samples. The authors should compare BET surface area, crystallinity and crystallite size for all materials. These results should be correlated with the activity of samples.
  4. Description of experimental section is not sufficient to provide reproducibility of the results. For instance, (page 3, verses: 114-115) “Then a certain amount of urea or niobium oxalate hydrate was put into the suspension under stirring for 2 h to get a uniform solution.” There is no information about concentration or assumed molar ratio of the reagents. Furthermore, there is no information about assumed concentration of nitrogen dopant in 10NNbO sample.
  5. Some claims are not supported by literature data. The authors should carefully read the text and add appropriate references when necessary. For instance, page 6, verses 232-233; Page 2, verses: 74-78; etc.
  6. Analysis and description of XPS data must be improved. Why binding energy of Nb 3d peaks in Nb2O5 sample is much higher than that reported in literature (approximately 207.1 eV)? Shift of O 1s and Nb 3d peaks in the same direction and by the similar value indicates that there may be some serious problem with normalization of XPS data. Please check it carefully.
  7. As concerns synthesis of materials, some of the materials were heated with different heating rate (5 or 10 °C/min). Why? What is the impact of this parameter on activity of the materials?
  8. According to the experimental section, the authors withdrawn more than 80% of initial reaction mixture in order to calculate conversion of the pollutant. Such high reduction of volume of the reaction mixture may have significant impact on reliability of the results. It is clearly seen in Fig. S4. According to this figure, for some of the samples the conversion increased, then maintained unchanged, decreased and increased again. In my opinion, there is some serious problem in experimental procedure used to evaluate photocatalytic activity of the samples. The authors must modify the experimental procedure and repeat these experiments.
  9. As concerns Raman spectroscopy, the authors wrote: “Also, a new peak appeared at 824 cm-1 demonstrated the formation of new chemical bond between N and Nb” (page 6, verses: 238-239). I would like to point out that the same Raman peak can be observed for samples without nitrogen.
  1. According the authors, all nitrogen-doped samples have almost the same nitrogen concentration. Why it is the case? What was the efficiency of nitrogen incorporation for all these materials? What is the reproducibility of the synthesis procedure?
  2. The authors wrote (page 6, verses 220-221): “The absence of obvious peaks of nitrogen element indicates that the introduction of a small amount of nitrogen does damage the lattice arrangement. While, the introduction of N causes the transition of the dominant crystal facet.”

Where the XRD peak typical of nitrogen is expected? What is expected origin of this peak?

  1. The insets in Fig. 1C are unclear and must be improved.

Reviewer 3 Report

My comments are in attached file.

Round 2

Reviewer 1 Report

The authors have missed 2 comments from old review comments

I) Page 70

This comment was not answered, "I understand that TiO2 system are exhibiting wide band gap, thus need UV rays for the process of Photocatalysis. "

The main concern is 
1) 3.6 eV of band gap is not narrow band gap, hence it need to be corrected
2) Efforts are important to identify/discover the photocatalysts  those are working in visible range (for energy less than or equal to 3.0eV)

---------

II) Page 366 - What is circle Hydrothermal method?

Reviewer 2 Report

In my opinion, the authors did their best to improve the manuscript. 

Author Response

Thank you very much for your recognition of our work and efforts. Thank you again for your advice and careful work.